# (GIGA)byte

DATA RELEASE

# Chromosomal-level genome assembly of the long-spined sea urchin *Diadema setosum* (Leske, 1778)

Hong Kong Biodiversity Genomics Consortium*,†

## ABSTRACT

The long-spined sea urchin *Diadema setosum* is an algal and coral feeder widely distributed in the Indo-Pacific that can cause severe bioerosion on the reef community. However, the lack of genomic information has hindered the study of its ecology and evolution. Here, we report the chromosomal-level genome (885.8 Mb) of the long-spined sea urchin *D. setosum* using a combination of PacBio long-read sequencing and Omni-C scaffolding technology. The assembled genome contains a scaffold N50 length of 38.3 Mb, 98.1% of complete BUSCO (Geno, metazoa_odb10) genes (the single copy score is 97.8% and the duplication score is 0.3%), and 98.6% of the sequences are anchored to 22 pseudo-molecules/chromosomes. A total of 27,478 gene models have were annotated, reaching a total of 28,414 transcripts, including 5,384 tRNA and 23,030 protein-coding genes. The high-quality genome of *D. setosum* presented here is a valuable resource for the ecological and evolutionary studies of this coral reef-associated sea urchin.

**Submitted:** 11 January 2024

\* Correspondence on behalf of the consortium. E-mail: jeromehui@cuhk.edu.hk

† Collaborative Authors: Entomological experts who validated the dataset and their affiliations appears at the end of the document

Preprint submitted at https://doi.org/10.1101/2024.01.16.575490

Included in the series: **Hong Kong Biodiversity Genomics** (https://doi.org/10.46471/GIGABYTE_SERIES_0006)

**Subjects** Genetics and Genomics, Animal Genetics, Marine Biology

## INTRODUCTION

Similar to other echinoderms, sea urchins lack a vertebral column and can metamorphose from juvenile bilateral swimming larvae into radial symmetrical adults [1, 2]. Owing to their critical phylogenetic position, sea urchins offer an understanding of how deuterostomes evolved [3–7]. To date, 21 sea urchin genomes of 16 species are available according to the data presented on NCBI; however, only nine of them are assembled at the chromosomal level: three in the order Temnopleuroida, including *Lytechinus variegatus* [8] and *Lytechinus pictus* [9], and six in the order Camarodonta, including *Heliocidaris erythrogramma* [10], *Heliocidaris tuberculata* [10], *Echinometra lucunter* [11], *Echinometra sp. EZ* [12], *Paracentrotus lividus* [13], and *Strongylocentrotus purpuratus* [14] (see the comparison table of different urchin genomes in figshare [15]).

*Diadema setosum* (Leske, 1778, NCBI:txid31175) in the order Diadematoida, commonly known as the porcupine or long-spined sea urchin, is considered one of the oldest known extant species in the genus *Diadema* [16]. *D. setosum* displays features of a typical sea urchin, including a dorso-ventrally compressed body equipped with particularly long, brittle, and hollow spines that are mildly venomous [17, 18]. This species can be easily differentiated from other *Diadema* species by the presence of five distinctive white dots at the aboral side around the anal pore between the ambulacral grooves (Figure 1A). Sexually matured individuals have been documented to have an average weight from 35 to 80 g and an average test size from 7 to 8 cm in diameter and approximately 4 cm in height [16, 19].

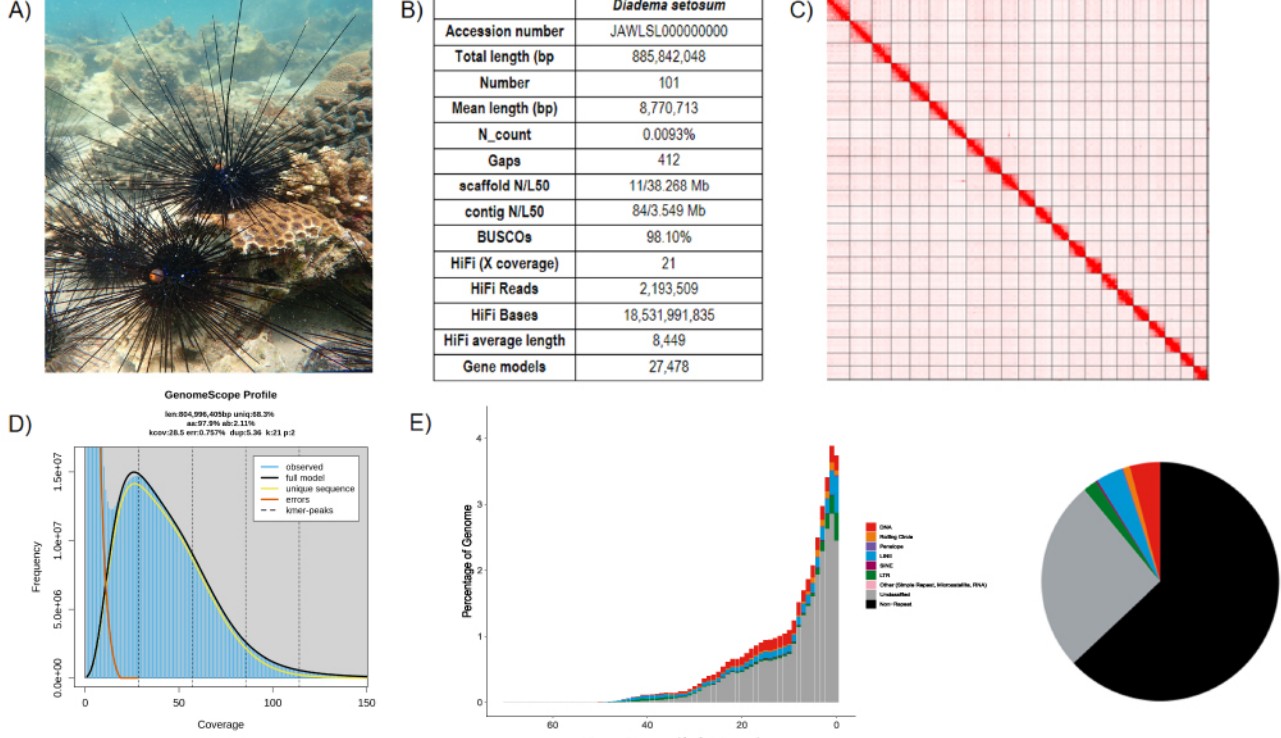

**Figure 1.** Genomic information of *Diadema setosum*.
(A) Photo of *D. setosum*; (B) Statistics of the assembled genome; (C) Omni-C contact map of the assembly visualised using Juicebox (v1.11.08, RRID:SCR_021172) (details can be found in Table 2); (D) Genomescope report with k-mer = 21; (E) Repetitive elements distribution in the assembled genome.

Due to its high invasiveness to localities beyond its natural range, *D. setosum* is now widely distributed in the tropical regions throughout the Indo-Pacific basin and can now be found latitudinally from Japan to Africa and longitudinally from the Red Sea to Australia [20].

*D. setosum* can thrive at depths of up to 70 m below sea level and is usually reef-associated [21]. It is a prolific grazer that feeds on the macroalgae that can be found on the surface of various substrata, as well as the algae that are associated with the coral skeleton [22, 23]. While a normal level of grazing eliminates competitive algae and can potentially offer a more suitable environment for coral settlement and development, overgrazing results in a reduction in coral community complexity, which in turn deteriorates the reef ecosystem and reduces the complexity of the coral community [24, 25]. Furthermore, overpopulated sea urchins can reduce coral recruitment and the growth of juvenile coral can be hindered [26–28]. Here, we present the chromosomal-level genome assembly of *D. setosum*. This valuable resource provides insights into the ecology and evolution of echinoderms, thereby enhancing further studies on sea urchins.

## CONTEXT

Here, we report a high-quality genome assembly of *D. setosum* in the order Diadematoida and family Diadematidae.

## METHODS

### Collection and husbandry of samples

The long-spined sea urchins, *D. setosum*, were collected at the coastal area of the Tolo Channel in Hong Kong (22.4872, 114.3082) in November 2022. The animals were maintained in 35 ppt artificial seawater at 23 °C until the DNA and RNA isolation, and fed with frozen clams or shrimps once a week.

### Isolation of high molecular weight genomic DNA, quantification, and qualification

High molecular weight (HMW) genomic DNA was isolated from a single individual. The urchin was first removed from the culture and the test was opened with a pair of scissors. The internal tissue, except the gut, was snap-frozen in liquid nitrogen and ground to fine powder. DNA extraction was performed with the Qiagen MagAttract HMW kit (Qiagen Cat. No. 67563) following the manufacturer's protocol. In brief, 1 g of powdered sample was put in a microcentrifuge tube with 200 µl 1× PBS. Subsequently, RNase A, Proteinase K, and Buffer AL were added to the tube. The mixture was incubated at room temperature (~22 °C) for 3 hours. The sample was then eluted with 120 µl of elution buffer (PacBio Ref. No. 101-633-500). Throughout the extraction progress, wide-bore tips were used whenever DNA was transferred. The eluted sample was quantified by the Qubit® Fluorometer, Qubit™ dsDNA HS, and BR Assay Kits (Invitrogen™ Cat. No. Q32851). In total, 10 µg of DNA was collected. The purity of the sample was evaluated by the NanoDrop™ One/OneC Microvolume UV–Vis Spectrophotometer, with the standard A260/A280: ~1.8 and A260/A230: >2.0. The quality and the fragment distribution of the isolated genomic DNA were examined by the overnight pulse-field gel electrophoresis, together with three DNA markers ($\lambda$-Hind III digest, Takara Cat. No. 3403; DL15,000 DNA Marker, Takara Cat. No. 3582A and CHEF DNA Size Standard-8-48 kb Ladder, Cat. No. 170-3707). The DNA was then diluted in elution buffer to prepare a 300 ng solution for gel electrophoresis. The electrophoresis profile was set as follows: 5k as the lower end and 100k as the higher end for the molecular weight; Gradient = 6.0 V/cm; Run time = 15 h:16 min; Included angle = 120°; Int. Sw. Tm = 22 s; Fin. Sw. Tm = 0.53 s; Ramping factor: a = Linear. The gel was run in 1.0% PFC agarose in 0.5× TBE buffer at 14 °C.

### DNA shearing, library preparation, and sequencing

A total of 10 µg of *D. setosum* DNA in 120 µl elution buffer was transferred to a g-tube (Covaris Part No. 520079). The sample was then centrifuged six times at 2,000 × g for 2 min. The sheared DNA was collected with a 2 ml DNA LoBind® Tube (Eppendorf Cat. No. 022431048) at 4 °C until the library preparation. Overnight pulse-field gel electrophoresis was performed to examine the fragment distribution of the sheared DNA, with the same electrophoresis profile described in the previous section.

A SMRTbell library was then constructed with the SMRTbell® prep kit 3.0 (PacBio Ref. No. 102-141-700) following the manufacturer's protocol. In brief, the sheared DNA was first subjected to DNA repair, and both ends of each DNA strand were polished and tailed with an A-overhang. Ligation of T-overhang SMRTbell adapters was then performed and the SMRTbell library was purified with SMRTbell® cleanup beads (PacBio Ref. No. 102158-300). The concentration and size of the library were examined with the Qubit® Fluorometer,

**Table 1.** Genome and transcriptome sequencing information.

| Genome sequencing data | | | | |
|---|---|---|---|---|
| Library | No. of reads | No. of bases | Coverage (X) | Accession |
| PacBio HiFi | 2,193,509 | 18,531,991,835 | 21 | SRR24631719 |
| Omnic | 450,451,192 | 67,567,678,800 | 76 | SRR26502301 |
| Transcriptome sequencing data | | | | |
| Sample name | No. of reads | No. of bases | Accession | |
| DseRNA | 40,875,262 | 6,131,231,889 | SRR24694066 | |

Qubit™ dsDNA HS, and BR Assay Kits (Invitrogen™ Cat. No. Q32851), and an overnight pulse-field gel electrophoresis, respectively. A nuclease treatment step was performed afterward to remove any non-SMRTbell structures in the library, and a final size-selection step was performed to remove the short fragments in the library with 35% AMPure PB beads.

The Sequel® II binding kit 3.2 (PacBio Ref. No. 102-194-100) was used for the final preparation of sequencing. In brief, Sequel II primer 3.2 and Sequel II DNA polymerase 2.2 were annealed and bound to the SMRTbell structures in the library. Then, the library was loaded at an on-plate concentration of 90 pM using the diffusion loading mode. The sequencing was performed on the Sequel IIe System with the internal control provided by the binding kit. The sequencing was prepared and run in 30-hour movies, with 120 min pre-extension. The movie was captured by the software SMRT Link v11.0 (PacBio) and HiFi reads were generated and collected for further analysis. In total, one SMRT cell was used in the sequencing. Details of the sequencing data are listed in Table 1.

## Omni-C library preparation and sequencing

An Omni-C library was constructed using the Dovetail® Omni-C® Library Preparation Kit (Dovetail Cat. No. 21005) according to the manufacturer's instructions. In brief, 60 mg of frozen powered tissue sample was added into 1 mL 1× PBS, where the genomic DNA was crosslinked with formaldehyde, and the DNA was then digested with endonuclease DNase I. Subsequently, the concentration and fragment size of the digested sample was validated by the Qubit® Fluorometer, Qubit™ dsDNA HS, and BR Assay Kits (Invitrogen™ Cat. No. Q32851), and the TapeStation D5000 HS ScreenTape, respectively. Afterwards, both ends of the DNA were polished and a biotinylated bridge adaptor was ligated at 22 °C for 30 min. Next, proximity ligation between crosslinked DNA fragments was performed at 22 °C for 1 hour, followed by the reverse crosslinking of DNA and its purification with SPRIselect™ Beads (Beckman Coulter Product No. B23317).

End repair and adapter ligation were performed with the Dovetail™ Library Module for Illumina (Dovetail Cat. No. 21004). In brief, DNA was tailed with an A-overhang and ligated with Illumina-compatible adapters at 20 °C for 15 min. The Omni-C library was then sheared into small fragments with USER Enzyme Mix and purified with SPRIselect™ Beads. Subsequently, DNA fragments were isolated with Streptavidin Beads. Universal and Index PCR Primers from the Dovetail™ Primer Set for Illumina (Dovetail Cat. No. 25005) were used to amplify the DNA library. A final size selection step was completed with SPRIselect™ Beads with DNA fragments ranging between 350 bp and 1000 bp only. The concentration and fragment size of the sequencing library were assessed by the Qubit® Fluorometer, Qubit™ dsDNA HS, and BR Assay Kits, and the TapeStation D5000 HS ScreenTape, respectively. The qualified library was sequenced on an Illumina HiSeq-PE150 platform. Details of the sequencing data are listed in Table 1.

## RNA extraction and transcriptome sequencing

Total RNA was extracted from the internal tissues of the same individual used for DNA extraction using TRIzol reagent (Invitrogen) following the manufacturer's protocol. The quality of the extracted RNA was validated with the NanoDrop™ One/OneC Microvolume UV–Vis Spectrophotometer (Thermo Scientific™ Cat. No. ND-ONE-W) and 1% agarose gel electrophoresis. The qualified samples were sent to Novogene Co. Ltd (Hong Kong, China) for the construction of a polyA-selected RNA sequencing library using the TruSeq RNA Sample Prep Kit v2 (Illumina Cat. No. RS-122-2001) and 150 bp paired-end sequencing. Agilent 2100 Bioanalyser (Agilent DNA 1000 Reagents) was used to measure the insert size and concentration of the final library. Details of the sequencing data are shown in Table 1.

## Genome assembly and gene model prediction

*De novo* genome assembly was completed using Hifiasm (RRID:SCR_021069) [29] with default parameters, and the Hifiasm output assembly was BLAST (RRID:SCR_004870) to the NT database, and the BLAST output was used as input for Blobtools (v1.1.1, RRID:SCR_017618) [30] to validate and remove any possible contaminations (Figure 2). Haplotypic duplications of the primary assembly were detected and removed using purge_dups (RRID:SCR_021173) according to the depth of HiFi reads [31] with default parameters. Proximity ligation data from the Omni-C library were used to scaffold the PacBio genome by YaHS [32]. A Kmer-based statistical analysis was used to estimate the heterozygosity, while the repeat content and size were analyzed by Jellyfish (RRID:SCR_005491) [33] and GenomeScope (RRID:SCR_017014) [34]. Transposable elements (TEs) were annotated using the automated Earl Grey TE annotation pipeline (version 1.2) [35]. The mitochondrial genome was assembled using MitoHiFi (v2.2) [36].

For gene model prediction, the RNA sequencing data was first processed with Trimmomatic (RRID:SCR_011848) [37] and transformed into transcripts using genome-guided Trinity (RRID:SCR_013048) [38]. Augustus (RRID:SCR_008417) [39] was trained using BUSCO (RRID:SCR_015008) [40], while GeneMark-ET (RRID:SCR_011930) [41] was used for *ab initio* gene prediction. Gene models were then predicted by funannotate (v1.8.5, RRID:SCR_023039) [42] using the parameters "--repeats2evm --protein_evidence uniprot_sprot.fasta --genemark_mode ET --optimize_augustus --organism other --max_intronlen 350000". The gene models from several prediction sources, including GeneMark, high-quality Augustus predictions, PASA (RRID:SCR_014656), Augustus, GlimmerHMM [43], and SNAP (RRID:SCR_007936), were passed to Evidence Modeler to generate the annotation files. PASA was employed to update the EVidenceModeler (EVM) consensus predictions [44]. In addition, untranslated region annotations were added, and models for alternatively spliced isoforms were created.

## DATA VALIDATION AND QUALITY CONTROL

Quality checks of samples during DNA extraction and PacBio library preparation were performed by NanoDrop™ One/OneC Microvolume UV–Vis Spectrophotometer, Qubit® Fluorometer, and overnight pulse-field gel electrophoresis. The Omni-C library was subjected to quality check by Qubit® Fluorometer and TapeStation D5000 HS ScreenTape.

For the genome assembly, the validation of contamination scaffolds from the Hifiasm output was done by searching the NT database through BLAST. The resulting output was analysed by BlobTools (v1.1.1) [32] (Figure 2). Furthermore, a Kmer-based statistical

blobtools_pacbio.blobDB.json.bestsum.phylum.p8.span.100.blobplot.bam0

**Figure 2.** Genome assembly quality control and contaminant/cobiont detection.

**Table 2.** GenomeScope statistics with K-mer length 21.

| Property | min | max |
|---|---|---|
| Homozygous (aa) | 97.85% | 97.93% |
| Heterozygous (ab) | 2.07% | 2.15% |
| Genome Haploid Length (bp) | 795,402,175 | 804,996,405 |
| Genome Repeat Length (bp) | 251,904,808 | 254,943,312 |
| Genome Unique Length (bp) | 543,497,367 | 550,053,093 |
| Model Fit | 75.49% | 98.83% |
| Read Error Rate | 0.76% | 0.76% |

approach was used to estimate the genome heterozygosity. The repeat content and their size were estimated by Jellyfish [33] and GenomeScope (Figure 1E and Table 2) [34]. BUSCO (v5.5.0) [40] was run to evaluate the completeness of the genome assembly and gene annotation with the metazoan dataset (metazoa_odb10).

## RESULTS

A total of 18.5 Gb of HiFi bases were generated with an average HiFi read length of 8,449 bp with 21× data coverage (Table 1). The assembled genome size was 885.8 Mb, with 101 scaffolds and a scaffold N50 of 38.3 Mb in 11 scaffolds, contig N50 of 3.5 Mb in 84 contigs, and a complete BUSCO estimation of 98.1% (the single copy score was 97.8% and the duplication score was 0.3%), (metazoa_odb10) (Figure 1B; Table 3). By incorporating 67.5 Gb Omni-C data, the assembly anchored 98.6% of the scaffolds into 22 pseudochromosomes, which matches the karyotype of *D. setosum* (2n = 44) [45] (Figure 1C; Table 4). The assembled *D. setosum* genome size is comparable to other published sea urchin genomes [8–11] and to the estimated size of 804 Mb by GenomeScope with a 2.11% heterozygosity rate (Figure 1D; Table 2). Moreover, telomeric repeats were identified in 16 out of 22 pseudochromosomes (Table 5).

Total RNA sequencing data was obtained from a single *D. setosum* individual. The final assembled transcriptome contained 135,063 transcripts, with 113,391 Trinity annotated genes (with an average length of 838 bp and a N50 length of 1,456 bp), and was used to perform gene model prediction. A total of 27,478 gene models were generated with 23,030 predicted protein-coding genes, with a mean coding sequence length of 483 amino acids (Figure 1B; Table 3).

For repeat elements, a total repetitive content of 36.98% was identified in the assembled genome, including 25.87% unclassified elements (Figure 1E; Table 6). Among the known repeats, DNA was the most abundant (4.18%), followed by long interspersed nuclear elements (3.64%) and long terminal repeats (1.92%). In contrast, Rolling Circle, short interspersed nuclear elements (SINE), Penelope, and others were only present in low proportions (Rolling Circle: 0.92%, SINE: 0.23%, Penelope: 0.17%, others: 0.04%).

## CONCLUSION AND FUTURE PERSPECTIVES

Sea urchin *D. setosum* (Diadematoida) belongs to a key phylogenetic group of animals in evolutionary history. This animal is characterised by deuterostomic development and is ecologically important to coral reefs. Prior to this study, there was a limited amount of high-quality sea urchin genomes, and the genomic resource for this ecologically important Diadematoida was missing. Here, we presented a high-quality chromosomal-level genome assembly of *D. setosum*, providing a valuable resource and foundation for a better understanding of the ecology and evolution of sea urchins.

**Table 3.** Genome assembly statistic and sequencing information.

| Species | *Diadema setosum* |
|---|---|
| Total_length | 885,842,048 |
| Number | 101 |
| Mean length (bp) | 8,770,713 |
| Longest | 52,803,307 |
| Shortest | 1,000 |
| N_count | 0.0093% |
| Gaps | 412 |
| N50 | 38,268,380 |
| N50n | 11 |
| N70 | 37,598,940 |
| N70n | 15 |
| N90 | 35,597,747 |
| N90n | 20 |
| BUSCOs (Genome, metazoa_odb10) | C:98.1%[S:97.8%,D:0.3%],F:1.2%,M:0.7%,n:954 |
| HiFi (X coverage) | 21 |
| HiFi Reads | 2,193,509 |
| HiFi Bases | 18,531,991,835 |
| HiFi Q30% | 2 |
| HiFi Q20% | 4 |
| HiFi GC% | 38 |
| HiFi Average length (bp) | 8,449 |
| Gene models | 27,478 |
| Number of protein-coding genes | 23,030 |
| BUSCOs (Proteome, metazoa_odb10) | C:95.5%[S:91.6%,D:3.9%],F:1.6%,M:2.9%,n:954 |
| Total length of protein-coding genes (AA) | 11,124,603 |
| Mean_length of protein-coding genes (AA) | 483 |

**Table 4.** Scaffold information of 22 pseudochromosomes.

| Chr Number | Scaffold_id | Scaffold_length | Sum % of the whole genome |
|---|---|---|---|
| 1 | scaffold_1 | 52,803,307 | 5.96% |
| 2 | scaffold_2 | 50,852,986 | 11.70% |
| 3 | scaffold_3 | 45,525,403 | 16.84% |
| 4 | scaffold_4 | 43,371,471 | 21.74% |
| 5 | scaffold_5 | 43,357,684 | 26.63% |
| 6 | scaffold_6 | 42,989,390 | 31.48% |
| 7 | scaffold_7 | 42,816,929 | 36.32% |
| 8 | scaffold_8 | 41,224,596 | 40.97% |
| 9 | scaffold_9 | 40,183,210 | 45.51% |
| 10 | scaffold_10 | 38,460,970 | 49.85% |
| 11 | scaffold_11 | 38,268,380 | 54.17% |
| 12 | scaffold_12 | 38,267,842 | 58.49% |
| 13 | scaffold_13 | 38,257,702 | 62.81% |
| 14 | scaffold_14 | 38,191,111 | 67.12% |
| 15 | scaffold_15 | 37,598,940 | 71.36% |
| 16 | scaffold_16 | 36,752,172 | 75.51% |
| 17 | scaffold_17 | 36,404,119 | 79.62% |
| 18 | scaffold_18 | 36,371,564 | 83.73% |
| 19 | scaffold_19 | 36,008,329 | 87.79% |
| 20 | scaffold_20 | 35,597,747 | 91.81% |
| 21 | scaffold_21 | 30,718,448 | 95.28% |
| 22 | scaffold_22 | 29,453,580 | 98.60% |

**Table 5.** List of the telomeric repeats identified in the genome.

| Scaffold ID | Strand | Position | Sequence |
|---|---|---|---|
| scaffold_1 | Reverse | end | TTAGGGGTTAGGGTTGGGTTAGAGGTTAGCGTTAAGGGTCTAAGGTTAGG |
| scaffold_2 | Reverse | end | AGGGTTAGGGTTAGGGTTAGGGTTAGGGTTAGGGTTAGGGGTTAGGTTAG |
| scaffold_3 | Reverse | end | AGGTTAGGGTTAGGGGTTAGGGTTAGGGTTAGGGTTAGGGGTTAGGGTTAG |
| scaffold_4 | Forward | start | CCCTAACCCTAACCCTAACCCTAACCCTAACCCTAACCCTAACCCTAACC |
| scaffold_4 | Reverse | end | GGTTAGGGGTTAGGGTTAGGGTTAGGGTTAGGGGTTAGGGTTAGGGTTAG |
| scaffold_5 | Forward | start | AACCCTAACCCTAACCCTAACCCTAACCCTAACCCTAACCCTAACCCTAA |
| scaffold_6 | Reverse | end | GGTTAGGGTTAGGGTTAGGGTTAGGGTTAGGGTTAGGGTTAGGTTAGGGT |
| scaffold_7 | Forward | start | CCTAACCCTAACCCAAACCCTAACCCTAACCCTAACCCTAACCCTACCCT |
| scaffold_7 | Reverse | end | GGGTTAGGGTTAGGGTTAGGGTTAGGGTTAGGGTTAGGGTTAGGGTTAGG |
| scaffold_9 | Reverse | end | GGGTTAGGGTTAGGGTTAGGGTTAGGGTTAGGGTTAGGGTTAGGGTTAGA |
| scaffold_10 | Forward | start | TAACCCTAACCCTAACCCTAACCTAACCCTAACCCTAACCCTAACCCTAA |
| scaffold_10 | Reverse | end | TTGGGTTAGGGTAGGGTTAGGGGTTTGGTTAGGGTTAGGGTTAGGGGTAG |
| scaffold_14 | Forward | start | CTAACCCTAACCCTAACCCCTAACCCTAACCCTAACCCTAACCCTAACCC |
| scaffold_14 | Reverse | end | AGGGTTAGGGTTAGGGGTTAGGGTTAGGGTTAGGGTTAGGGTTAGGTTAG |
| scaffold_15 | Forward | start | TAACCCTAACCCTAACCCTAACCCTAACCCTAACCCTAACCCTAACCCTA |
| scaffold_16 | Forward | start | CTAACCCTAACCCCTAACCCTAACCCTAACCCTAACCCTAACCCTAACCC |
| scaffold_17 | Forward | start | AACCCTAACTACCTAACCCTTAACTCCTAACCCTAACTCCTTAACCCTAT |
| scaffold_19 | Reverse | end | GTTAGGGTTAGGGTTAGGGTTAGGGTTAGGGTTAGGGTTAGGGTTAGGGT |
| scaffold_20 | Reverse | end | GGGTTAAGGTTAGGGTTAGGGTTAGGGTTAGGGTTAGGGGGTTAGGGTTA |
| scaffold_21 | Reverse | end | AGGGTTAGGGTTAGGGTTAGGGTTAGGGTTAGGGTTAGGGTTAGGGTTAG |

**Table 6.** Statistics of the annotated repetitive elements.

| Classification | Total length (bp) | Count | Proportion (%) | No. of distinct classifications |
|---|---|---|---|---|
| DNA | 37,064,214 | 103,475 | 4.18 | 8,730 |
| LINE | 32,263,932 | 51,458 | 3.64 | 7,860 |
| LTR | 16,983,326 | 16,349 | 1.92 | 4,119 |
| Other (Simple Repeat, Microsatellite, RNA) | 345,889 | 1,007 | 0.04 | 586 |
| Penelope | 1,546,508 | 3,266 | 0.17 | 1,530 |
| Rolling Circle | 8,121,340 | 11,509 | 0.92 | 2,468 |
| SINE | 2,065,090 | 7,829 | 0.23 | 758 |
| Unclassified | 229,187,329 | 386,068 | 25.87 | 8,808 |
| **SUM** | **327,577,628** | **580,961** | **36.98** | **34,859** |

## DATA AVAILABILITY

The final genome assembly was deposited to NCBI with the accession number GCA_033980235.1. The raw reads generated in this study were submitted to the NCBI database under the BioProject accession PRJNA973839. The genome and genome annotation files, as well as additional tables and additional Information were also deposited in Figshare [15].

## ABBREVIATIONS

HMW, High molecular weight; SINE, short interspersed nuclear element; TE, transposable element.

## DECLARATIONS

## Ethics approval and consent to participate

The authors declare that ethical approval was not required for this type of research.



## Competing interests

The authors declare that they do not have competing interests.

## Authors' contribution

JHLH, TFC, LLC, SGC, CCC, JKHF, JDG, SCKL, YHS, CKCW, KYLY, and YW conceived and supervised the study. APYC and THWF collected the sea urchin samples. HYY maintained the animal culture. WLS performed DNA extraction, library preparation, and genome sequencing. WN carried out the genome assembly and gene model prediction.

## Funding

This work was funded and supported by the Hong Kong Research Grant Council Collaborative Research Fund (C4015-20EF), CUHK Strategic Seed Funding for Collaborative Research Scheme (3133356), and CUHK Group Research Scheme (3110154).

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

## DETAILS OF COLLABORATIVE AUTHORS

### • List of authors in Hong Kong Biodiversity Genomics Consortium

Jerome H. L. Hui,[1] Ting Fung Chan,[2] Leo Lai Chan,[3] Siu Gin Cheung,[4] Chi Chiu Cheang,[5,6] James Kar-Hei Fang,[7] Juan Diego Gaitan-Espitia,[8] Stanley Chun Kwan Lau,[9] Yik Hei Sung,[10,11] Chris Kong Chu Wong,[12] Kevin Yuk-Lap Yip,[13,14] Yingying Wei,[15] Wai Lok So,[1] Wenyan Nong,[1] Apple Pui Yi Chui,[16] Thomas Hei Wut Fong,[16] Ho Yin Yip[1]

[1]School of Life Sciences, Simon F.S. Li Marine Science Laboratory, State Key Laboratory of Agrobiotechnology, Institute of Environment, Energy and Sustainability, The Chinese University of Hong Kong, Hong Kong, China

[2]School of Life Sciences, State Key Laboratory of Agrobiotechnology, The Chinese University of Hong Kong, Hong Kong SAR, China

[3]State Key Laboratory of Marine Pollution and Department of Biomedical Sciences, City University of Hong Kong, Hong Kong SAR, China

[4]State Key Laboratory of Marine Pollution and Department of Chemistry, City University of Hong Kong, Hong Kong SAR, China

[5]Department of Science and Environmental Studies, The Education University of Hong Kong, Hong Kong SAR, China

[6]EcoEdu PEI, Charlottetown, PE, C1A 4B7, Canada

[7]Department of Food Science and Nutrition, Research Institute for Future Food, and State Key Laboratory of Marine Pollution, The Hong Kong Polytechnic University, Hong Kong SAR, China

[8]The Swire Institute of Marine Science and School of Biological Sciences, The University of Hong Kong, Hong Kong SAR, China

[9]Department of Ocean Science, The Hong Kong University of Science and Technology, Hong Kong SAR, China

[10]Science Unit, Lingnan University, Hong Kong SAR, China

[11]School of Allied Health Sciences, University of Suffolk, Ipswich, IP4 1QJ, UK

[12]Croucher Institute for Environmental Sciences, and Department of Biology, Hong Kong Baptist University, Hong Kong SAR, China

[13]Department of Computer Science and Engineering, The Chinese University of Hong Kong, Hong Kong SAR, China

[14]Sanford Burnham Prebys Medical Discovery Institute, La Jolla, CA, USA

[15]Department of Statistics, The Chinese University of Hong Kong, Hong Kong SAR, China

[16]School of Life Sciences, The Chinese University of Hong Kong, Hong Kong SAR, China

