## [Editor Report]

Editor’s AssessmentThis work is part of a series of papers from the Hong Kong Biodiversity Genomics Consortium sequencing the rich biodiversity of species in Hong Kong. This example assembles the genome of the long-spined sea urchin Diadema setosum (Leske, 1778). Using PacBio HiFi long-reads and Omni-C data the assembled genome size was 886 Mb, consistent to the size of other sea urchin genomes. The assembly anchored to 22 pseudo-molecules/chromosomes, and a total of 27,478 genes including 23,030 protein-coding genes were annotated. Peer review added more to the conclusion and future perspectives. The data hopefully providing a valuable resource and foundation for a better understanding of the ecology and evolution of sea urchins.

---

## [Reviewer Report]

Reviewer name and names of any other individual's who aided in reviewer Remi KetchumDo you understand and agree to our policy of having open and named reviews, and having your review included with the published papers. (If no, please inform the editor that you cannot review this manuscript.)YesIs the language of sufficient quality?YesPlease add additional comments on language quality to clarify if needed
Are all data available and do they match the descriptions in the paper? YesAdditional CommentsAre the data and metadata consistent with relevant minimum information or reporting standards? See GigaDB checklists for examples <a href="http://gigadb.org/site/guide" target="_blank">http://gigadb.org/site/guide</a>YesAdditional CommentsIs the data acquisition clear, complete and methodologically sound?YesAdditional CommentsIs there sufficient detail in the methods and data-processing steps to allow reproduction?YesAdditional CommentsIs there sufficient data validation and statistical analyses of data quality? YesAdditional CommentsIs the validation suitable for this type of data?YesAdditional CommentsIs there sufficient information for others to reuse this dataset or integrate it with other data?YesAdditional CommentsAny Additional Overall Comments to the AuthorMinor Edits  Line 62: Change to “lack a vertebral column” instead of “lack the” Line 64: Change to “sea urchins” instead of sea urchin Line 70: Ketchum et al 2022 in GBE produced a chromosome-level genome assembly of Echinometra sp. EZ so this citation should be included here.  Line 91: change to “results in a reduction in coral community complexity”   I think that the end of the introduction could use a sentence or two that explicitly states why this genome will be a valuable resource to the scientific community. I think this will also help wrap up the introduction.   Line 101: Can you provide coordinates? Also could you remove the word ‘alive.’ Line 130: I am confused by what you mean “the sample was then proceeded”  Line 181: Was this the same individual that you used for genomic DNA isolation? Line 196: please could you include the specific flags that you used for purge_dups? Did you run Hifiasm with the default parameters?  Line 240: I would definitely try and include some more sentences in this section.  Line 253: Is this section supposed to be here? I think this is meant to go into the methods section.   The authors could think about potentially a comparison table of the different urchin genome stats that are available currently? I would also encourage the readers to generate KAT plots to validate that they have successfully collapsed the haplotypes – a common problem with higher heterozygosity. 
RecommendationMinor Revision

---

## [Reviewer Report]

Reviewer name and names of any other individual's who aided in reviewer F. MarlétazDo you understand and agree to our policy of having open and named reviews, and having your review included with the published papers. (If no, please inform the editor that you cannot review this manuscript.)YesIs the language of sufficient quality?YesPlease add additional comments on language quality to clarify if needed
Are all data available and do they match the descriptions in the paper? YesAdditional CommentsAre the data and metadata consistent with relevant minimum information or reporting standards? See GigaDB checklists for examples <a href="http://gigadb.org/site/guide" target="_blank">http://gigadb.org/site/guide</a>YesAdditional CommentsIs the data acquisition clear, complete and methodologically sound?YesAdditional CommentsIs there sufficient detail in the methods and data-processing steps to allow reproduction?YesAdditional CommentsIs there sufficient data validation and statistical analyses of data quality? YesAdditional CommentsIs the validation suitable for this type of data?YesAdditional CommentsIs there sufficient information for others to reuse this dataset or integrate it with other data?YesAdditional CommentsAny Additional Overall Comments to the AuthorI think it would be great to give further detail on the statistics out of the hifiasm contiging step. What are the contig statistics (after the hifiasm step)? RecommendationAccept

---

## [Reviewer Report]

Reviewer name and names of any other individual's who aided in reviewer Phillip DavidsonDo you understand and agree to our policy of having open and named reviews, and having your review included with the published papers. (If no, please inform the editor that you cannot review this manuscript.)YesIs the language of sufficient quality?YesPlease add additional comments on language quality to clarify if needed
Minor language errors that should be corrected in copy-editingAre all data available and do they match the descriptions in the paper? YesAdditional CommentsAre the data and metadata consistent with relevant minimum information or reporting standards? See GigaDB checklists for examples <a href="http://gigadb.org/site/guide" target="_blank">http://gigadb.org/site/guide</a>YesAdditional CommentsIs the data acquisition clear, complete and methodologically sound?YesAdditional CommentsIs there sufficient detail in the methods and data-processing steps to allow reproduction?YesAdditional CommentsIs there sufficient data validation and statistical analyses of data quality? YesAdditional CommentsIs the validation suitable for this type of data?YesAdditional CommentsIs there sufficient information for others to reuse this dataset or integrate it with other data?YesAdditional CommentsAny Additional Overall Comments to the AuthorIn their work, Hui et al present a chromosome-level genome assembly for Diadema setosum, the long-spined urchin. This new data is especially exciting given no high-quality genomic resource for the Diadematoida is available, bolstering comparative genomics work of echinoderms and the study of this species. Overall, the methods and data are well described and have produced a high quality genome assembly and associated annotations that will be a valuable addition to the community. I have a handful of primarily minor suggestions detailed below:  Major comments:  1. Conclusions and future perspectives: Currently, this section is only a sentence and states the new assembly will “further understanding of ecology and evolution of sea urchins”, which I think is a little uninspiring. I think more detail can be provided in this section to explain how this genome assembly adds to current knowledge. For example, reiterating that this is the first chromosome-level Diadematoida assembly, or perhaps explaining with examples how a good reference genome can inform ecological studies. Overall, the significance of this work is not really explained which I think sells this nice work short.  Minor comments:   1. Lines 232-233 state the mean coding sequence is 483 bp which seems a bit low, but having examined the peptide fasta file, I believe the average amino acid length is 483 AA, giving an average coding sequence length of ~1449bp. Please confirm and correct if necessary. This would also increase the total # of coding basepairs listed in Table 1.   2. Lines 66-71: The authors state there are 5 chromosome-level sea urchin assemblies, all of which are camarodonts. However, I believe there are at least three additional chromosome-level assemblies for sea urchins not mentioned: 1) Echinometra sp. EZ (Ketchum et al, 2022; https://academic.oup.com/gbe/article/14/10/evac144/6717576 ) and 2) Paracentrotus lividus (Marletaz et al, 2023; https://www.sciencedirect.com/science/article/pii/S2666979X23000617?via%3Dihub ) and 3) Strongylocentrotus purpuratus (https://www.echinobase.org/echinobase/) Further, P. lividus is not a camarodont, so the text should be corrected accordingly.   3. Lines 106: Please state whether the individual samples for genome sequencing was male or female  4. Lines 54-54: The BUSCO score is reported at 98.1% but it should be be specified if this is the complete BUSCO score or the single-copy BUSCO score. Ideally, the single copy and duplication scores, rather than the complete, score is reported so readers have an idea for the duplication rate/haploid-ness of the assembly. Same issue on lines 221. Thank you for reporting in Table 1.   5. Line 56: Text states “27,478 genes including 23,030 protein coding genes” were annotated. Augustus often outputs genes and transcripts, so I am wondering if the authors mean 27K transcripts including 23K genes. If so, the authors should clarify. If not, I think a brief statement of what these additional 4K genes are would be informative  6. Table 1: Please clarify if “HiFi (X): 21” is referring to 21X coverage. Please correct length of coding sequence to amino acid sequence, and total coding sequence length. Same with Figure 1 panel B. 
RecommendationMinor Revision